# Hydrologic regimes drive nitrate export behavior in human impacted watersheds

Galen Gorski[1], Margaret A. Zimmer[1]

[1]Department of Earth and Planetary Sciences, University of California Santa Cruz, Santa Cruz, CA, 95064, United States

*Correspondence to*: Galen Gorski (ggorski@ucsc.edu)

**Abstract.** Agricultural watersheds are significant contributors to downstream nutrient excess issues. The timing and magnitude of nutrient mobilization in these watersheds are driven by a combination of anthropogenic, hydrologic, and biogeochemical factors that operate across a range of spatial and temporal scales. However, how, when, and where these complex factors drive nutrient mobilization has previously been difficult to capture with low-frequency or spatially limited datasets. To address this knowledge gap, we analyzed daily nitrate concentration (c) and discharge (Q) data for a four-year period (2016-2019) from five nested, agricultural watersheds in the midwestern United States that contribute nutrient loads to the Gulf of Mexico. These records allow us to investigate nutrient mobilization patterns at a temporal and spatial resolution not previously possible. The watersheds span two distinct landforms shaped by differences in glacial history resulting in natural soil properties that necessitated different drainage infrastructure across the study area. To investigate nutrient export patterns under different hydrologic conditions, we partitioned the hydrograph into stormflow and baseflow periods and examined those periods separately through the analysis of their concentration-discharge (c-Q) relationships on annual, seasonal, and event time scales. Stormflow showed consistent chemostatic patterns across all seasons, while baseflow showed seasonally dynamic c-Q patterns. Baseflow exhibited chemodyanmic conditions in the summer and fall and more chemostatic conditions in the winter and spring, suggesting that water source contributions during baseflow were nonstationary. Baseflow chemodynamic behavior was driven by low-flow, low-$NO_3^-$ conditions during which in-stream and near-stream biological processing likely moderated in-stream $NO_3^-$ concentrations. Additionally, inputs from deeper groundwater with longer residence times and lower $NO_3^-$ concentration likely contributed to low-$NO_3^-$ conditions in-stream, particularly in the larger watersheds. Stormflow c-Q behavior was consistent across watersheds, but baseflow c-Q behavior was linked to intensity of agriculture and density of built drainage infrastructure, with more drainage infrastructure associated with higher loads and more chemostatic export patterns across the watersheds. This suggests that how humans 'replumb' the subsurface in response to geologic conditions has implications for hydrologic connectivity, homogenization of source areas, and subsequently nutrient export during both baseflow and stormflow. Our analysis also showed that anomalous flow periods greatly influenced overall c-Q patterns, suggesting that the analysis of high-resolution records at multiple scales is critical when interpreting seasonal or annual patterns.

## 1 Introduction

Excess nutrient export to streams can have detrimental effects on human health and ecosystem function, by contaminating drinking water (Weyer et al., 2001) and contributing to harmful algal blooms (Howarth, 2008), hypoxia (Jenny et al., 2016) and loss of species diversity in receiving water bodies (Diaz and Rosenberg, 2008). Globally, the number of hypoxic dead zones that have been identified in the scientific literature has roughly doubled each decade, now reaching well over 500 (Conley et al., 2011). The spatial extent and severity of dead zones are often correlated to temporal patterns in upstream nitrogen loading from contributing catchments (Rabalais et al., 2009; Turner et al., 2012).

One of the largest dead zones in the world is in the northern Gulf of Mexico, which experiences expansive eutrophication each spring and summer due to nutrient export from largely agricultural watersheds within the Upper Mississippi River Basin (Rabalais et al., 2002). In response, many US states have invested considerable resources in developing nutrient reduction strategies with the goal of mitigating nutrient mobilization and downstream effects. For example, the Iowa Nutrient Reduction Strategy has a goal of reducing nitrogen loads in Iowa streams by 45%, committing $560 million to meet that goal in 2019 alone (Iowa State University, 2020).

Despite the considerable investments in developing solutions, downstream water bodies still receive substantial nitrogen loading from their upstream watersheds (Bouraoui and Grizzetti, 2011; Sprague et al., 2011). One reason for this persistence is the build-up of excess applied nitrogen that can remain in the subsurface for decades and contribute to in-stream nitrate ($NO_3^-$) loads long after application practices have changed, or mitigations strategies have been implemented (Fovet et al., 2015; Sebilo et al., 2013). These, and other $NO_3^-$ sources create a heterogeneous patchwork of source areas throughout the landscape that can become "activated" or "deactivated" in response to changing hydrologic conditions (Abbott et al., 2018; Dupas et al., 2019). A better understanding of what factors contribute to source area activation, and the timing of their activation is critical to predicting in-stream $NO_3^-$ concentrations and loads and ultimately developing operational nutrient management strategies.

The examination of the relationship between solute concentration and stream discharge (c-Q relationships), in combination with other information about watershed structure and land use practices, can be an effective way to investigate contributing source zones within a watershed (e.g. Godsey, 2009; Thompson et al., 2011). When viewed in log-log space, solute concentration and discharge often vary linearly according to a slope, which can be used to describe the relative tendency of a watershed to transport or retain the solute under various hydrologic conditions (Basu et al., 2010; Musolff et al., 2017). Slopes near zero (|c-Q slope| ≤ 0.2) indicate chemostatic behavior in which solute concentration varies little in response to changes in discharge. Chemostatic conditions can arise when contributing areas have uniform solute concentrations, as is often seen with $NO_3^-$ in areas with intensive agriculture (Bieroza et al., 2018; Thompson et al., 2011). In contrast, chemodynamic behavior is characterized by slopes different from zero in which the solute concentration is sensitive to changes in discharge. Chemodynamic conditions can arise from source areas with more heterogeneous solute concentrations which may become activated under different hydrologic conditions (Dupas et al., 2019). The c-Q relationship can be characterized as an enrichment pattern if the slope is positive (c-Q slope > 0.2) or a dilution pattern if the slope is negative (c-Q slope < -0.2).

Recent studies have recognized that c-Q relationships vary as a function of flow percentile, suggesting that the structure of hydrologic connectivity is driven by flow conditions (Diamond and Cohen, 2018; Jones et al., 2017; Zimmer et al., 2019). Recently, the accessibility of data from high-frequency sensor networks has allowed the exploration of these relationships at a time scale previously difficult to observe. For example, high-frequency datasets have been used to investigate c-Q behavior at the event scale, revealing dynamic changes in $NO_3^-$ sourcing and processing at short timescales (Blaen et al., 2017; Bowes et al., 2015; Carey et al., 2014; Kincaid et al., 2020). However, much previous work has focused on a single catchment, and/or data collected over a relatively short period of time. This makes it difficult to determine how the connections between the hydrologic, biogeochemical, and anthropogenic factors, which operate over a range of temporal and spatial scales, influence in-stream $NO_3^-$ concentrations. For example, antecedent moisture conditions, and precipitation timing and intensity reflect changes that occur over hours or days (Rozemeijer et al., 2010), while vegetation dynamics, and on-farm practices such as crop planting and fertilization reflect seasonal changes (Minaudo et al., 2019; Royer et al., 2006). Additionally, the influence of these factors are impacted by differences in watershed-specific characteristics, such as topography, soil type, land use practices, and geologic history (Marinos et al., 2020; Moatar et al., 2017; Wymore et al., 2017). Understanding how these processes and watershed characteristics interact across the relevant spatial and temporal scales in heavily managed watersheds is a crucial step in developing strategies to mitigate downstream impact (Hansen et al., 2018).

Only recently have high-resolution records become sufficiently long and instrumentation sufficiently wide-spread to examine c-Q relationships under different hydrologic conditions in multiple locations. These higher resolution records allow us to examine nutrient mobilization patterns at the event, seasonal, and annual scale across key spatial gradients in a way not previously possible. Here, we analyze four years of publicly available daily measurements of discharge and $NO_3^-$ concentration from five nested agricultural watersheds in the midwestern United States. Using a semi-autonomous event picking algorithm, we partition the hydrograph into stormflow and baseflow periods to address the following research questions:

1) How do c-Q relationships during stormflow and baseflow periods vary by season, and what can that tell us about changes in hydrologic connectivity and nitrogen sources throughout the year?
2) What relationship do $NO_3^-$ concentration, load measurements, and c-Q relationships have to underlying and human-impacted watershed properties?
3) How can high-frequency records be used to identify distinct export regimes and characterize anomalous events that might play a disproportionate role in watershed c-Q behavior?

## 2 Methods

### 2.1 Site description

The Raccoon River watershed drains 8,870 km$^2$ of low-relief, heavily agricultural area in central Iowa, USA, which drains into the Gulf of Mexico (Figure 1). It is made up of the North Raccoon River watershed (USGS HUC: 07100007) and the South Raccoon River watershed (USGS HUC: 07100006).

For this study we subdivided the Raccoon River watershed into a series of five nested watersheds shown in Figure 1; the Upstream Sac City (*USC*) and the Middle Redfield (*MRF*) on the North Raccoon River, the Upstream Panora (*UPN*) on the Middle Raccoon River, the Middle Jefferson (*MJF*) on the South Raccoon River, and the Downstream Van Meter (*DVM*), which is below the confluence of the three major tributaries draining the area. The *MJF* is inclusive of *USC*; *MRF* is inclusive of *UPN*, and *DVM* is inclusive of the entire Raccoon River watershed. Typical of this area, agricultural productivity is the dominant land use in all five watersheds ranging from 85-92% of land use (Table S2), the vast majority of which is corn (*Zea mays L.)* and soybeans (*Glycine max L.)*.

The Raccoon River watershed is marked by a stark divide in landforms driven by recent glaciations, with the majority of the area underlain by glacial sediments deposited by the Des Moines Lobe during the last glaciation of the region approximately 12,000 years ago (Prior, 1991). These areas are characterized by poorly developed surface drainage networks and ephemeral surface water bodies. As a result, extensive tile drainages, ditches, and canals have been installed and constructed beginning as early as the 1800s to drain excess water from the subsurface (Figure 1). The southwestern portion of the Raccoon River watershed lies within the Southern Iowa Drift Plain, an area that was shaped by 500,000-year-old glacial advances that extended south into present day Missouri (Prior, 1991). This portion of the watershed is characterized by steeper topography and more naturally well-developed drainage networks, which require less drainage infrastructure such as tile drains, ditches, and canals. *UPN*, *MRF*, and *DVM* drain areas that overlay both the Des Moines Lobe and the Southern Iowa Drift Plain, while *USC and MJF* are entirely within the Des Moines Lobe.

The Raccoon River watershed is characterized by cold dry winters and warm wet summers, with an average annual precipitation of 850 mm (1981-2010;PRISM Climate Group, 2004), the majority of which falls as rain between April and October, aligning with the growing season.

**2.2 Datasets**

We analyzed in situ mean daily $NO_3^-$ concentration (c) and discharge (Q) data from the outlet of each watershed at gaging stations maintained by the U.S. Geologic Survey for *USC* (05482300), *MRF* (05483600), *MJF* (05482500), and *DVM* (05484500), and from the Iowa Institute of Hydraulic Research (IIHR) for *UPN* (WQS0032). To retrieve data, we used the dataRetrieval package in R (v 3.6.0) through the National Water Information System (De Cicco et al., 2018). Data for *UPN* was obtained directly from the IIHR. We analyzed daily discharge and $NO_3^-$ concentration data from January 2016 to December 2019, during which discharge records were complete for all sites and $NO_3^-$ records had > 88% coverage for all sites except *UPN*, which had 72% coverage (Table S1). At each gaging station, $NO_3^-$ concentrations were measured at 15-minute resolution (5-min for *UPN*) using Hach Nitratax plus sc probes (Hach, Loveland, CO) and aggregated to daily average $NO_3^-$ concentration for this study. $NO_3^-$ concentration averages are not volume weighted in an effort to facilitate comparison to maximum contaminant levels and target concentrations. Concentrations are measured as $[NO_3^-]+[NO_2^-]$ in mg/L as nitrogen, however because $[NO_2^-]$ are generally negligible, we refer to sensor measurements as "$NO_3^-$" concentration throughout.

To analyze land use characteristics for each watershed, we downloaded land use data from the National Land Cover Database 2016 at a 30 m x 30 m resolution (Dewitz, 2019). Land use data were binned into four categories;

water/wetlands, developed, forested/barren/shrubs, and crops (including pasture). Data for landforms, drainage infrastructure, and stream network were downloaded from the Iowa Department of Natural Resources. Digital elevation model (DEM) data were downloaded from AWS Open Terrain Tiles using the elevatr package (v 0.3.1) in R (Hollister et al., 2020). We downloaded daily precipitation data for the four-year period of analysis (2016-2019) for two sites (USC00137312 and USC00136566) within the Raccoon River watershed from the NOAA National Centers

for Environmental Information.

### 2.3 Event identification

We separated the discharge time series into baseflow and stormflow periods through semi-automating storm event identification using the following criteria: 1) $dQ/dt \geq$ 1e-4 cfs/second for the rising limb of the event, 2) $\max(Q_{event}) \geq 0.01*\max(Q_{record})$, and 3) the event duration $\geq$ 3 days. The end of each event was determined when either

the event falling limb $dQ/dt \geq 0$ or discharge returned to pre-event levels. For some, such as events that showed up as shoulder peaks on larger events, or those with indistinct peaks, visual inspection and subjective decisions were required (Figure S1). The criteria were derived from similar studies (Dupas et al., 2016; Knapp et al., 2020; Rozemeijer et al., 2010), and exact thresholds for were tuned and adapted for the structure and dynamics of the watersheds' hydrographs to ensure the selection of peaks. We tried several different approaches to identify events (including %

flow change) but found that the above criteria produced the most reasonable results. Because the watersheds are in close proximity and have generally similar characteristics, their hydrographs exhibit reasonably similar structure, which may make this method more suited to this type of analysis. If the analysis were expanded to include watersheds with very different hydrograph structures, a different approach may be merited. Time periods identified as storm events were classified as stormflow, and all other times were classified as baseflow (Figure 2).

We note that this classification scheme differs from traditional baseflow separation techniques that use graphical, geochemical, or isotopic approaches to identify and separate the proportion of the hydrograph that is comprised by baseflow and stormflow (Hooper and Shoemaker, 1986; Klaus and McDonnell, 2013). Baseflow separation techniques have shown that a large fraction of event water is derived from baseflow (e.g. Schilling & Zhang, 2004). Our goal is not to contradict or supplant this finding, but rather to illustrate how a simple partitioning of the

hydrograph based on peaks in discharge allows us to isolate nutrient export dynamics in specific hydrologic regimes.

### 2.4 Characterizing export regimes

Export patterns (chemostatic, dilution, or enrichment) were calculated for stormflow, baseflow, and the full record (herein referred to as stormflow+baseflow) for the full period of analysis and on a seasonal basis.

Concentration-discharge relationships for baseflow and stormflow+baseflow periods were calculated by aggregating data for the time period of interest. Stormflow c-Q relationships were calculated in two ways; first by aggregating data from all stormflow events over the time period of interest, and second, by calculating c-Q relationships for each individual storm event and averaging those values over all events (Figure S4). The former is referred to as bulk stormflow c-Q relationships, and the later as event-averaged c-Q relationships.

Seasonal and annual calculations were made based on the water year which begins on October 1st, and the year was divided seasonally into fall (October, November, December), winter (January, February, March), spring (April, May, June), and summer (July, August, September).

        We calculated the coefficient of variation (CV) for c and Q, and calculated the ratio of $CV_c$:$CV_Q$ to assess the relative variability of each (Musolff et al. 2015a; Thompson et al. 2011). $CV_c$:$CV_Q$ was calculated as:


$$\frac{CV_c}{CV_Q} = \frac{\mu_Q}{\mu_c} \frac{\sigma_c}{\sigma_Q} \qquad (1)$$

where μ represents the mean and $\sigma$ represents the standard deviation.

**2.5 Load estimations**

Cumulative $NO_3^-$ load estimates were calculated for each hydrologic regime (stormflow, baseflow, stormflow+baseflow) on an annual and seasonal basis as:

$$\sum_{i=1}^{n} c_i Q_i / f \qquad (2)$$

where $c_i$ and $Q_i$ are the daily $NO_3^-$ concentration and discharge values, and $f$ is the fraction of data coverage for the period of interest. If data were missing during a period, baseflow and stormflow loads were calculated based on their fractional contribution during the periods with data. All annual periods had $f > 0.75$, but some seasonal periods had low coverage, for seasonal periods where $f \leq 0.75$ no load estimate was calculated.

        Correlations between nutrient export parameters (load estimates and c-Q slopes) and landscape parameters were calculated using the Pearson correlation coefficient and significance was determined as $p < 0.05$ (Table S5).

**3 Results and discussion**
**3.1 Stream flow exhibits strong seasonality**

        In all five watersheds, 44-52% of the analysis period was classified as stormflow, with an average of 15
unique storm events in each watershed per year (Table 1). While the proportion of stormflow periods was similar between watersheds, the fraction of flow that was partitioned into stormflow and baseflow varied considerably between watersheds. *MJF* and *USC* had the highest proportion of stormflow, with 77.0 and 73.4% of annual flow classified as stormflow, respectively, compared to 62.4 and 63.9% in *UPN* and *MRF*, respectively (Table 1). This observation is consistent with the higher density of drainage infrastructure (e.g. canals, tile drainage) in *MJF* and *USC*,
leading to quicker routing of high flows to the stream channel compared to more natural drainage networks in *UPN* and *MRF*.

        Flow in all watersheds exhibited strong seasonality, with an average of 42.9% of total flow delivered in the spring. Summer months contributed the least to overall flow with an average of 17.3% across all watersheds. Despite differences in overall flow between the seasons, spring and summer experienced a similar number of stormflow events
across all watersheds (average of 5.5 in spring and 4.4 in summer), and similar precipitation totals (average 309 mm

in spring and 381 mm in summer). Increased streamflow in the spring months is likely a result of snow melt, rain on snow events, which can produce excess runoff, and increased crop growth in the summer months leading to more water retention.

### 3.2 $NO_3^-$ concentrations are sensitive to watershed characteristics, season, and hydrologic regime

The heavily tile-drained *USC* watershed showed the highest median $NO_3^-$ concentration of 9.23±3.09 mg/L (median ± standard deviation), while *MRF*, which has the least drainage infrastructure, showed the lowest (6.96±2.51 mg/L; Table S3). This is consistent with observations of increased stream $NO_3^-$ concentrations at the outlets of heavily tile-drained Iowa watersheds compared to those with less built drainage infrastructure (Schilling et al., 2012). The outlet of the the largest watershed, *DVM*, which receives contributions from *USC* and *MRF* showed intermediate $NO_3^-$ concentrations of 7.38±3.07 mg/L.

$NO_3^-$ concentrations displayed pronounced seasonality during stormflow and baseflow across all watersheds. Summer baseflow $NO_3^-$ concentrations showed a general decreasing trend with watershed area as the outlet of the largest watershed experienced the lowest concentration (Figure 3A). Low $NO_3^-$ concentrations in summer are often associated with lower flow periods which may have increased contributions from groundwater flow paths with longer residence times, and more streambed-water interaction. The lower flow and longer flow paths would allow for more nitrate processing in the subsurface and hyporheic zone, both have been positively associated with watershed area (Peralta-Tapia et al., 2015). In addition, summer periods have warmer temperatures, which promote biological nitrogen uptake activity (e.g. denitrification and assimilation) that can lower $NO_3^-$ concentrations (Moatar et al., 2017; Rode et al., 2016). Weakened correlations between baseflow $NO_3^-$ concentrations and watershed area during the rest of the year suggest that other processes may be more dominant at driving $NO_3^-$ concentrations during non-summer periods.

Maximum $NO_3^-$ concentrations were observed in the spring during both baseflow and stormflow periods (Figure 3). During stormflow periods, $NO_3^-$ concentrations correlated positively with drainage infrastructure density during all seasons, but the correlation was strongest during the spring months when the $NO_3^-$ concentrations were highest (Figure 3B). During spring precipitation events, water infiltrates rapidly through relatively bare soils, encountering accumulated nitrogen stocks in the shallow subsurface from previous years or early season fertilizer application and is routed off the landscape through tile drains (Van Meter et al., 2020; Royer et al., 2006). High flow periods can also reduce the ability of biological processes to alter $NO_3^-$ concentrations (Rodríguez-Blanco et al., 2015; Royer et al., 2006; Wollheim et al., 2018). This seasonality of $NO_3^-$ concentration has been previously observed in the Raccoon River watershed (Schilling and Zhang, 2004), as well as other agricultural catchments in the Midwest (Dupas et al., 2017; Van Meter et al., 2020; Pellerin et al., 2014).

### 3.3 Baseflow c-Q patterns reveal seasonally shifting nitrate processing and sources

Concentration-discharge relationships showed a difference between baseflow and bulk stormflow periods with baseflow periods exhibiting generally more chemodynamic c-Q slopes (Figure 4). Enriching chemodynamic export patterns (c-Q slope > 0.2) were observed during baseflow periods in all watersheds annually, with *UPN* showing

the strongest enrichment signal (c-Q slope = 0.79) and *USC* showing the weakest (c-Q slope = 0.21) (Figure 4A). Baseflow c-Q slopes were seasonally dynamic. Fall and summer showed generally higher c-Q slopes (blue and red triangles, respectively; Figure 4A), and winter and spring c-Q slopes were closer to zero (green and yellow triangles, respectively; Figure 4A).

There is a negative correlation between seasonal baseflow c-Q slope and drainage infrastructure density, which is strongest during the spring months ($R^2 = 0.85$; $p < 0.05$; Table S5). During these months, positive baseflow c-Q slopes are driven by low flow, low $NO_3^-$ concentration periods, which are less prevalent in the watersheds with a higher density of drainage infrastructure (*USC, MJF*, and *DVM*). The lack of low $NO_3^-$ concentration periods in these watersheds results in chemostatic c-Q slopes as the built drainage infrastructure serves to homogenize baseflow sources.

These human impacts can be highlighted by comparing the two end member watersheds in our dataset. *MRF*, which has the lowest density of drainage infrastructure ($0.37$ km/km$^2$), experienced chemodynamic enriching c-Q slopes across all seasons during baseflow, ranging from 0.34 in the winter to 0.75 in the summer. This suggests highly heterogenous source regions contributed to baseflow throughout the year. In contrast, *USC*, which has the highest density of drainage infrastructure ($1.11$ km/km$^2$) experienced chemodynamic conditions only in the summer (c-Q slope = 0.29), and chemostatic conditions across the other seasons. This suggests there were consistent, homogeneous sources producing stable $NO_3^-$ concentrations across a range of flow conditions throughout the year.

The strongest chemodynamic enrichment patterns occurred in the summer across all watersheds, while the most chemostatic season was generally the spring (Figure 4a). This pattern is exemplified in *DVM*, which integrates the signal from the other four upstream watersheds. The summer baseflow period in *DVM* is strongly enriching (c-Q slope = 0.75), while in spring, baseflow is chemostatic (c-Q slope = 0.08). This dynamic shift is driven by differences in baseflow $NO_3^-$ concentrations from spring to summer, suggesting differences in the sourcing or internal processing of baseflow from one season to the next (Richardson et al., 2020).

**3.4 Stormflow c-Q patterns show stationarity in seasonal $NO_3^-$ sources**

Bulk stormflow periods generally exhibited more chemostatic behavior than baseflow periods (Figure 4). The observation that low flow periods were more chemodynamic than high flow periods is consistent with other studies that have partitioned the hydrograph seasonally (Ehrhardt et al., 2019), by breakpoint analysis (Marinos et al., 2020), or by median discharge (Moatar et al., 2017), suggesting that this is a general feature of watershed hydrologic routing. There is considerable overlap in c and Q values between stormflow and baseflow periods (Figure S3). Given that baseflow and stormflow c-Q patterns differ, this suggests that partitioning of the hydrograph by events may sample different hydrologic regimes with similar discharges.

Bulk stormflow c-Q slope exhibited subtle seasonality with a slight dilution trend in winter c-Q slopes in several watersheds, and a slight enrichment trend in spring and summer (Figure 4B). Fall bulk stormflow c-Q slopes were chemostatic to weakly chemodynamic for all watersheds except *UPN*, which showed a c-Q slope of 0.71. This higher c-Q slope was driven by two anomalous, low $NO_3^-$ concentration events discussed in further detail in Section 3.5.

Although tile drained watersheds show higher stormflow $NO_3^-$ concentrations (Figure 3B), there does not appear to be a systematic effect on stormflow c-Q slopes (Figure 4B). This indicates that the nitrate sources activated during stormflow periods are transport-limited across all watersheds. That is, regardless of season, storms contribute flow to streams generally through shallow, quick flow paths that intersect high-$NO_3^-$ stores in these agriculturally intensive landscapes (Buda and DeWalle, 2009; Mellander et al., 2012).

Analysis of individual storm events reveals that event-averaged c-Q slopes form a narrow distribution around zero across all seasons (Figure 5A). Although many individual events could be classified as strongly chemodynamic if considered in isolation, examining the events in aggregate shows that there is a tendency towards chemostatic behavior across all watersheds (Figure 5B). The comparison of bulk stormflow c-Q slopes (Figure 4B) and event averaged c-Q slopes (Figure 5B) highlights the importance of c-Q event analysis at multiple temporal scales. If, for

example, three events each showed a chemostatic response but at different $NO_3^-$ concentration, they could be interpreted as chemodynamic when grouped together. Both methods of analysis could be useful in determining the nutrient export behavior of stormflow events which has been observed to be highly non-linear and hysteretic (Carey et al., 2014; Lloyd et al., 2016).

    Changes in c-Q slopes can be driven by changes in concentration, discharge or asymmetric changes in both

quantities. Comparison of $CV_c$ and $CV_Q$ values between baseflow and stormflow shows that baseflow c-Q chemodynamic behavior is driven by both a decrease in Q variation *and* an increase in c variation (Table S4). The ratio of $CV_c$:$CV_Q$ is higher during baseflow, consistent with variable sourcing of nitrate during these periods. During stormflow periods, $CV_c$:$CV_Q$ values are lower, indicating little change in c relative to Q. These patterns are more pronounced in the watersheds with the least amount of drainage infrastructure (*UPN* and *MRF*) than for the other

watersheds.

**3.5 Periods of anomalous flow and $NO_3^-$ concentrations can alter overarching riverine c-Q characteristics**

    During baseflow and stormflow periods, episodes of anomalous flow and $NO_3^-$ concentrations had a significant effect on c-Q slope analysis. In *UPN*, two events, during low flow periods in October 2017, had low $NO_3^-$

concentrations (average 1.34 and 0.46 mg/L; 4th and 2nd $NO_3^-$ concentration percentile across the whole study period, respectively). Individually, the events had c-Q slopes of -0.50 and 0.45. Inclusion of these events in the calculation of fall bulk stormflow c-Q behavior resulted in fall bulk stormflow c-Q slope of 0.71 (Figure 4B). However, with the removal of these events, the same calculation yields a slope of 0.09, much more in line with the other watersheds for the fall season. These events were included in our analysis, as they met the criteria for event selection, however their

ability to skew the bulk analysis is notable as they represent < 1% of annual flow and $NO_3^-$ load.

    Similarly, during baseflow in *MJF*, a period of anomalously low flow (mean = 96 cfs; < 0.1 flow percentile) and low nitrate concentration (mean = 0.05 mg/L; < 0.1 $NO_3^-$ concentration percentile) from 07/26/2017-10/19/2017 had a dramatic impact on the baseflow c-Q relationship (Figure S3D). Inclusion of the data from this period resulted in an annual baseflow c-Q slope of 1.42, indicating very strong enrichment behavior. Removal of the data from this

anomalous period decreased the slope to 0.42. Data from this time period may be influenced by biofouling, as such we do not include this period in further discussion of nutrient export behavior, but we do include it in our estimates of

annual and seasonal nitrate load, though it has little effect on our overall load estimates as the amount of nutrient export during this period is low. This is the only extended period of anomalous $NO_3^-$ concentrations that we have identified in these records, but care should be taken to identify anomalous and potentially erroneous periods when interpreting in situ records.

The ability of a single anomalous period (whether real or due to sensor artifacts) to influence the overall characterization of a hydrologic system highlights the difficulty of representing nutrient export behavior based on a single parameter fit across several seasons and flow regimes (Diamond and Cohen, 2018; Dupas et al., 2017; Fazekas et al., 2020; Marinos et al., 2020). This also highlights the need for high-frequency data collection activities that allow researchers and water quality practitioners to observe anomalous events during periods of the year that are not traditionally targeted by discrete or synoptic sampling campaigns (Wymore et al., 2019).

Periods of anomalous flow and $NO_3^-$ concentration highlight the year-to-year variability inherent in these systems (Jones et al., 2017). The records examined in this study do not cover a sufficient length of time to thoroughly examine inter-annual variability, and it remains an open question as to how stable the c-Q patterns presented in this study are over time. However, robust patterns such as seasonal non-stationarity in baseflow (Figure 4), and individual event c-Q patterns trending toward chemostatis (Figure 5) are likely to persist year after year.

**3.6 Seasonal patterns in nitrate load across watersheds**

Annual average $NO_3^-$ export across the study watersheds ranged from 4216±768 kg-N/km$^2$/yr in *USC* to 2222±371 kg-N/km$^2$/yr in *MRF*. Partitioning the hydrograph into seasonal stormflow and baseflow periods allows the identification of periods which contribute disproportionately to annual watershed $NO_3^-$ export magnitudes (Figure 6).

Spring stormflow periods accounted for the largest contribution to annual load across all watersheds, with an average of 37.5±11.5% for all years. Spring stormflow contributions displayed a large spatiotemporal range, from 19.7% in *UPN* in 2016 to 59.8% in *DVM* in 2017. Summer stormflow loads also showed considerable variation, with an average contribution of 9.4% of annual load, but ranging from < 1% (19.4 kg-N/km$^2$/yr) in 2017 to 18.3% (711 kg-N/km$^2$/yr) in 2018.

These ranges in $NO_3^-$ loads are largely driven by observed variation in summer stormflow events. For example, in the summer of 2017, which had an anomalously low $NO_3^-$ load, there were fewer stormflow events than average. Specifically, there was an average of 1.8 events across the watersheds with zero events identified in *USC* and *MJF*. In contrast, there was an average of 6.0 events across all five watersheds in summer 2018, which has anomalously high nitrate load. Additionally, the identified events in summer 2017 were approximately 22% the size of the events in summer 2018. This variability highlights the difficulty in predicting loads across seasons, hydrologic regimes and watersheds.

Baseflow loads showed considerable seasonal variability, although they consistently made up ≤ 15% of the annual load in each watershed. Baseflow loads typically peaked in the spring months, likely due to a seasonally high water table, which increased shallow groundwater contribution to streams (Jiang et al., 2010; Molenat et al., 2008). Additionally, spring fertilizer application and plowing can increase surface leaching, increasing the nitrate pool in the

shallow subsurface (Royer et al., 2006). That said, there were some discrepancies within individual watersheds; *UPN* had the highest seasonal baseflow export in fall and *MRF* had similar fall and spring baseflow loads (Figure 6A).


### 3.7 Nutrient export is driven by the spatial distribution of land use types and hydrologic infrastructure

There is a systematic trend toward higher $NO_3^-$ load in watersheds with a higher density of built drainage infrastructure (Figure 7), consistent with other studies (Basu et al., 2010; Musolff et al., 2015; Schilling and Zhang, 2004). The slope of the relationship between $NO_3^-$ load and drainage infrastructure density is much shallower for

baseflow than for stormflow, given the greater range in observed stormflow load across the watersheds (Kennedy et al., 2012). Drainage structures and tile drains route water from high $NO_3^-$ source areas directly to the stream, decreasing travel time and bypassing riparian areas that are highly active in nutrient processing (Dosskey et al., 2010). These structures are common features in agricultural landscapes and show strong correlation to the amount of cropped area across the five watersheds analyzed ($R^2 = 0.95$; $p < 0.01$).

The short circuiting of subsurface flow paths and increased cropped area drives watershed nutrient export patterns towards chemostatic behavior by homogenizing the source regions and limiting nutrient cycling during transport (Marinos et al., 2020; Musolff et al., 2015; Thompson et al., 2011). These patterns are most clear during both baseflow and stormflow periods in the spring months, when tile drains likely have their greatest influence on hydrologic routing. During the spring months, both baseflow and stormflow $NO_3^-$ loads are strongly correlated with

drainage infrastructure density ($R^2 = 0.88$ and 0.88; $p < 0.05$ and $p < 0.05$, respectively; Table S5). Additionally, export regimes are chemostatic (average c-Q slope = 0.15 for stormflow and 0.18 for baseflow).

In contrast, summer baseflow periods showed the strongest chemodynamic enrichment patterns with an average c-Q slope of 0.73 across all watersheds. The baseflow $NO_3^-$ load during the summer is most strongly correlated with the percentage of cropped area within 100 m of the stream ($R^2 = 0.94$; $p < 0.01$; Table S5). This suggests that

summer chemodynamic regimes are driven by low flow, low $NO_3^-$ periods where source areas that are proximal to the stream are contributing more significantly to discharge (Molenat et al., 2008). Lower density of agricultural activity in riparian areas (Table S2) leads to more heterogeneous source regions, which promotes low $NO_3^-$ load and the observed chemodynamic behavior. Other landscape factors such as drainage density or network topology likely influence summer baseflow loads as well, however, the strong correlation coupled with independent analysis in other

watersheds suggest that land use in riparian areas exhibits strong influence on baseflow nitrate loads (Wherry et al., 2021).

Seasonal and annual c-Q slopes across all hydrologic regimes show only weak correlations with watershed area suggesting that drainage infrastructure and the distribution and intensity of agriculture are the dominant drivers of $NO_3^-$ export regime in these watersheds. This is consistent with a recent study of 33 agricultural watersheds in the

Midwest (Marinos et al., 2020). Our results show that both conditions that lead to high $NO_3^-$ loads, whether hydrologic (i.e. stormflow) or landscape (i.e. increases in drainage infrastructure and agricultural intensity) are associated with chemostatic behavior. This trend is in line with the idea that landscapes with such agricultural intensity are a saturated solute source, whose delivery is flow-limited (Thompson et al., 2011).

**4 Conclusions**

Detailed analysis of event, seasonal, and annual $NO_3^-$ export showed that all five heavily agricultural watersheds showed similar temporal patterns of $NO_3^-$ load with highs in spring stormflow and lows in summer baseflow. Stormflow across all seasons was largely chemostatic and spring stormflow accounted for ~40% of annual loads. In contrast, baseflow periods exhibited seasonality in export regimes, with low summer flows driving periods of chemodynamic enrichment and winter and spring driving more chemostatic behavior in the winter and spring. The differences in c-Q behavior between stormflow and baseflow suggests that the systems dynamically, but predictably, shift between $NO_3^-$ export patterns in response to hydrologic forcing. There was a systematic trend toward more chemostatic behavior and higher $NO_3^-$ loads with increasing density of drainage infrastructure and agricultural land use across the five watersheds. These anthropogenic controls on $NO_3^-$ export in these watersheds are driven by disparate glacial histories across the watersheds that necessitate different flow routing infrastructure. During baseflow conditions, land use near the stream has a large impact on $NO_3^-$ loads, indicating that buffer strips or other near-stream management practices may be effective management practices for reducing loads during these periods.

Analysis of specific low-flow periods demonstrated that anomalous periods have the power to significantly affect our classification of export patterns and influence our understanding of watersheds as a whole. This highlights the dynamic nature of these systems and argues for event, seasonal, and longer-term analyses of nutrient export, particularly when attempting to measure the efficacy of management practices such as reductions in fertilizer application or near-stream buffer strips. High-resolution hydrochemical observations allow the detailed characterization of storm events which facilitate more accurate estimates of $NO_3^-$ loads than have been previously measured using regression-based techniques with sparse sample resolution. This study demonstrates the utility of high spatial and temporal resolution water quality sampling to disentangle the key factors controlling watershed nutrient export as well as the important role of state and federal water quality monitoring programs in addressing important water quality issues.

**Data availability**

Records that have been portioned into stormflow and baseflow for this analysis can be found at http://www.hydroshare.org/resource/173cff98da3c4263a110cba8c6d62406.

**Author contribution**

GG and MZ designed the study, GG carried out the analysis. GG and MZ prepared the manuscript.

**Competing interests**

The authors declare that they have no conflict of interest.

**Acknowledgements**

The authors would like to thank the Iowa Institute of Hydraulic Research for generously providing data. We also thank one anonymous reviewer and Doug Burns for their thoughtful feedback and suggestions during the review process.

**Funding**

This work was supported by the National Science Foundation Graduate Research Fellowship Program.

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

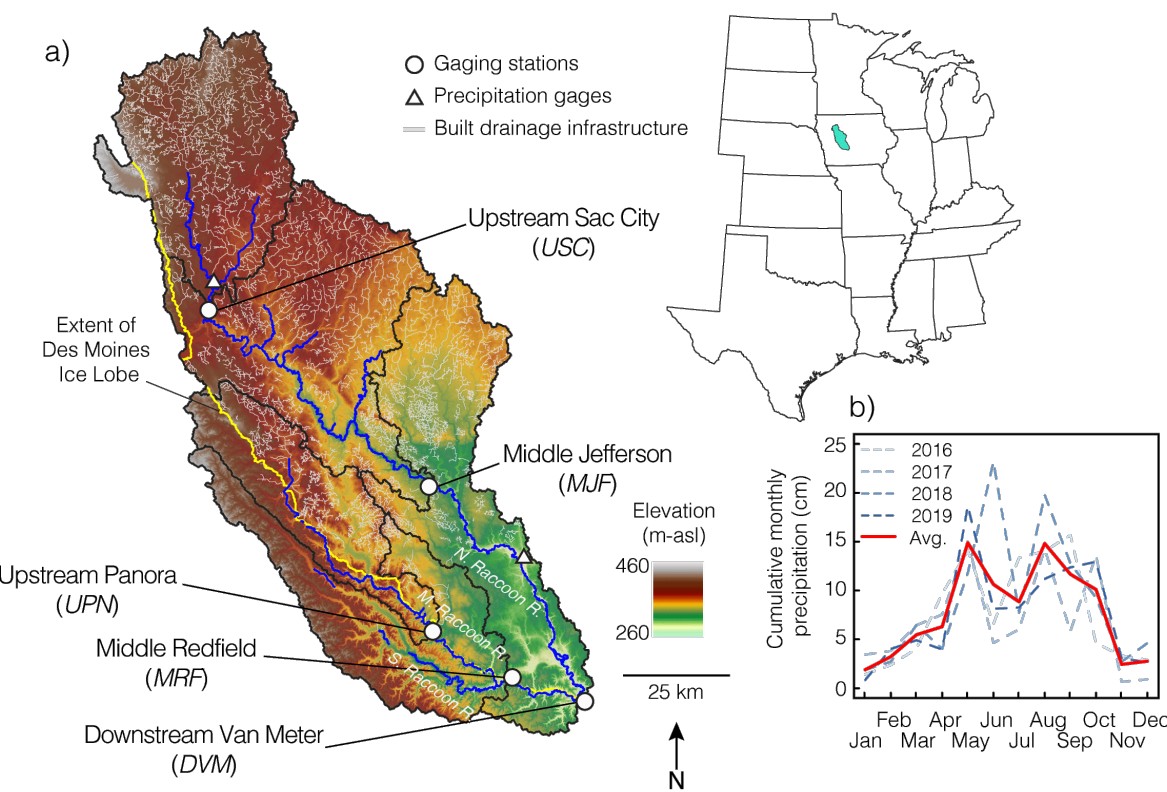

**Figure 1: A) Map of five watersheds (black outlines) analyzed in central Iowa along the North, Middle and South Raccoon Rivers. *MJF* is inclusive of *USC*, *MRF* is inclusive of *UPN*, and *DVM* is inclusive of the entire watershed pictured. The**
**yellow line maps the extent of the Des Moines Lobe in the last glaciation. Areas to the southwest of the line lie in the Southern Iowa Drift Plain. Built drainage infrastructure is shown in gray. Gaging stations (white circles) are along the North and Middle Raccoon Rivers (blue lines), the *DVM* gaging station is below the confluence of the branches of the Raccoon Rivers. Two precipitation gages are shown with white triangles. Precipitation data were averaged on a monthly basis across the four-year study period (2016-2019) and shown in (B); the red line indicates the monthly averages across the four years.**
**Landform and drainage infrastructure data were downloaded from the Iowa Department of Natural Resources.**

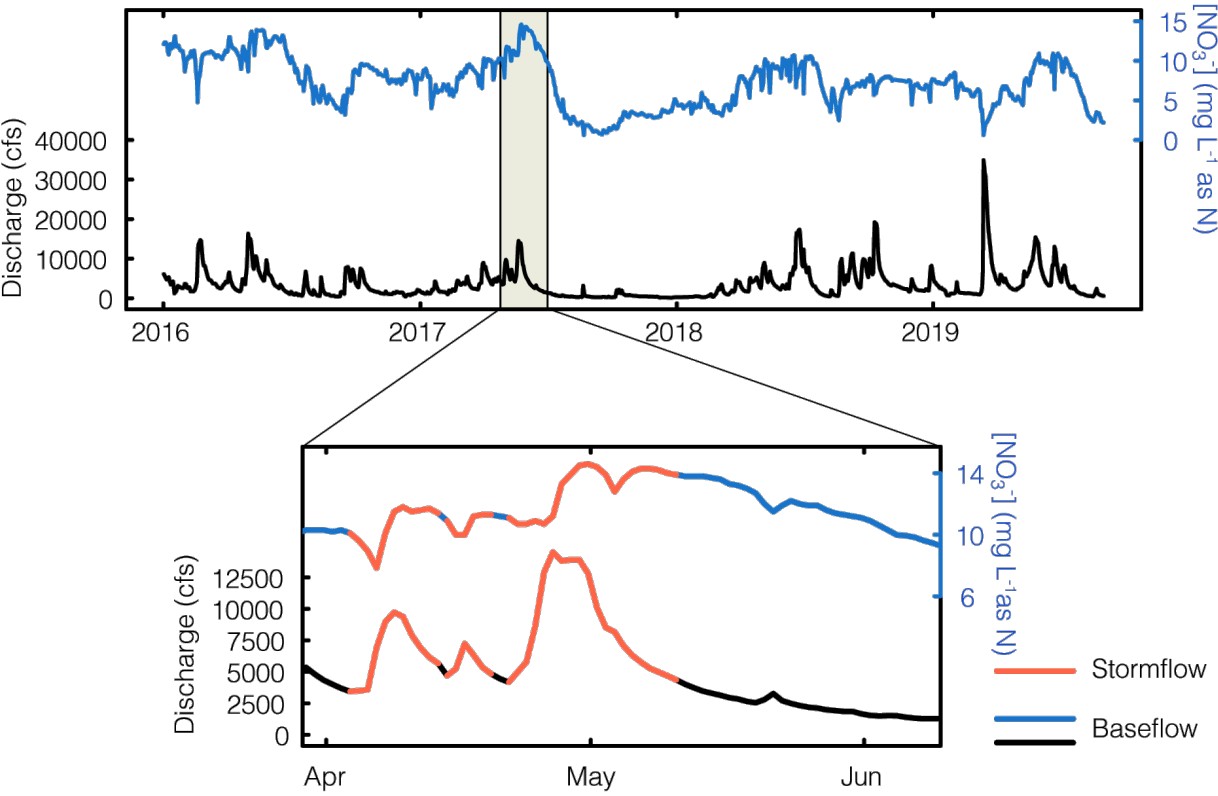

**Figure 2: Example hydrograph and chemograph from one gaging station over the four-year period of analysis, with expanded portion showing stormflow periods (red), and baseflow periods (blue and black). For full records of all five watersheds see Figure S1.**

**Table 1: Watershed hydrologic characteristics**

| Watershed | Area | Stormflow discharge[a] | Baseflow discharge[a] | Stormflow events | OND | JFM | AMJ | JAS |
|-----------|------|------------------------|-----------------------|------------------|-----|-----|-----|-----|
| | (km$^2$) | (%) | (%) | (events yr$^{-1}$) | (events yr$^{-1}$) | (events yr$^{-1}$) | (events yr$^{-1}$) | (events yr$^{-1}$) |
| *UPN* | 1116 | 62.4 | 32.3 | 14.3 | 3.3 | 1.3 | 5.0 | 5.0 |
| | | (9.3) | (5.5) | (1.7) | (1.5) | (0.6) | (0.8) | (2.2) |
| *USC* | 1840 | 73.4 | 26.6 | 15.0 | 3.0 | 3.0 | 5.8 | 3.3 |
| | | (3.5) | (3.6) | (3.6) | (1.4) | (1.8) | (2.5) | (2.8) |
| *MRF* | 2548 | 63.3 | 36.8 | 16.8 | 3.8 | 2.3 | 5.0 | 5.8 |
| | | (4.9) | (4.1) | (1.3) | (1.0) | (1.0) | (0.8) | (1.7) |
| *MJF* | 4188 | 77 | 23.1 | 14.3 | 3.7 | 2.3 | 5.5 | 3.8 |
| | | (6.1) | (5.3) | (3.9) | (1.6) | (1.9) | (1.3) | (3.3) |
| *DVM* | 8870 | 72.9 | 27.1 | 15.8 | 3.0 | 2.3 | 6.3 | 4.3 |
| | | (6.7) | (4.6) | (5.1) | (1.9) | (1.0) | (1.3) | (2.8) |

[a] as a percent of total annual discharge, standard deviations are reported in parentheses


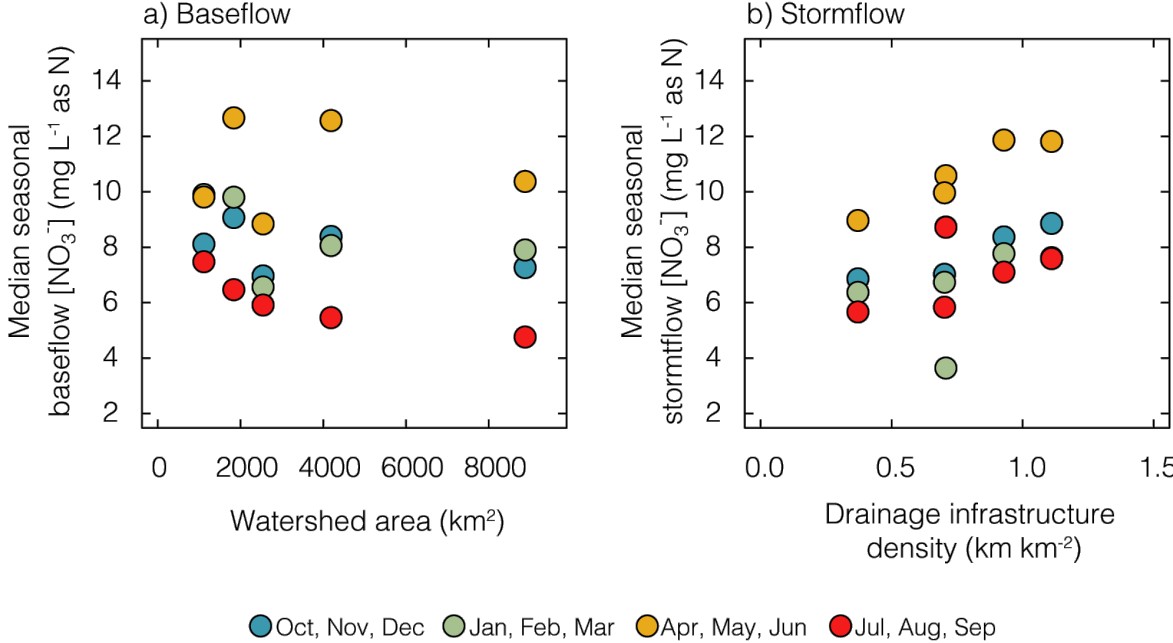

● Oct, Nov, Dec  ● Jan, Feb, Mar  ● Apr, May, Jun  ● Jul, Aug, Sep

**Figure 3: Seasonal median NO₃⁻ concentration during baseflow periods plotted against watershed area (A) and NO₃⁻**
**concentration during stormflow periods plotted against drainage infrastructure density (B). Baseflow NO₃⁻ concentration**
**showed the strongest correlation with watershed area during the summer months (red), and stormflow NO₃⁻ concentration**
**correlated the strongest with drainage infrastructure density during the spring months (yellow).**

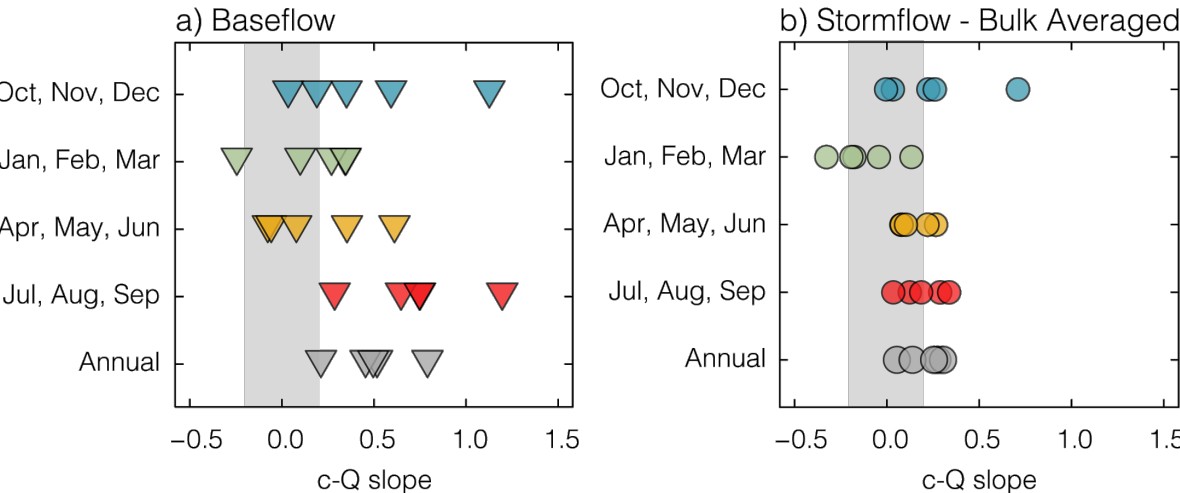

**Figure 4: Concentration-discharge slopes in each watershed calculated independently for baseflow (A), and bulk stormflow (B) for each season and annually (gray). Gray boxes indicate chemostatic behavior (|c-Q slope| ≤ 0.2).**

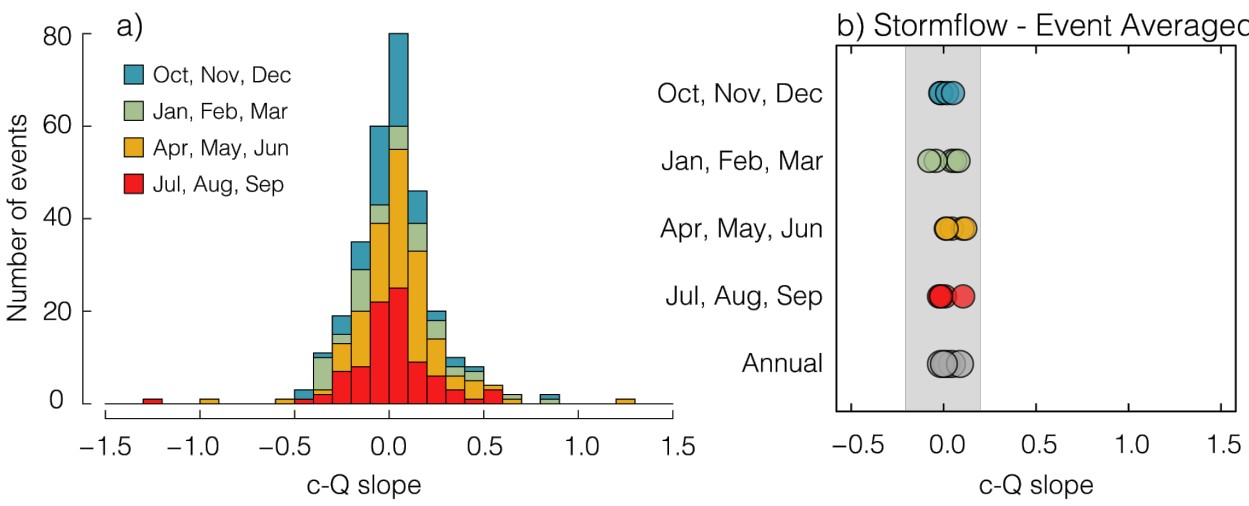


**Figure 5: (A) Individual storm event c-Q slopes for all five watersheds colored by season, and (B) event averaged stormflow c-Q slope by season calculated individually for each watershed, gray boxes indicate chemostatic behavior (|c-Q slope| ≤ 0.2).**



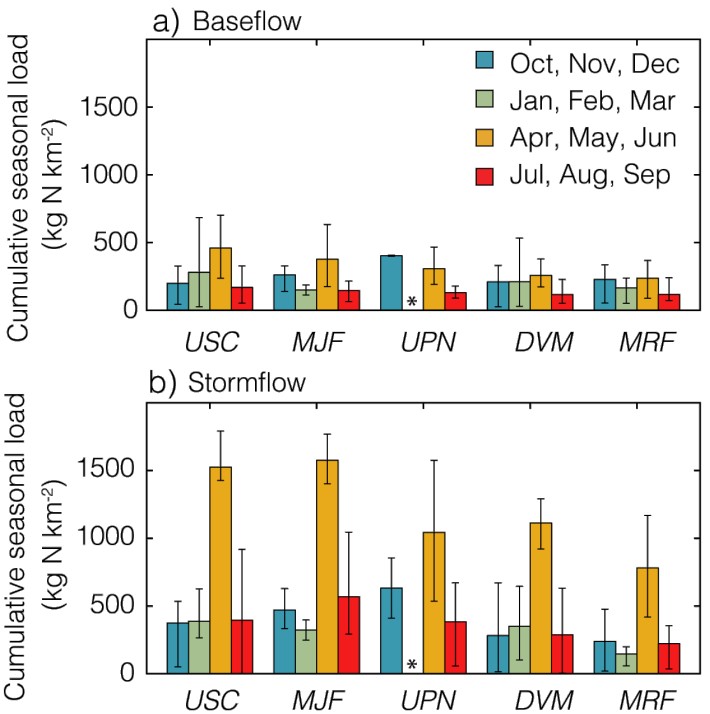

**Figure 6: Seasonal NO₃⁻ load (kg-N/km²) normalized by watershed area, averaged over the four years of analysis (2016-2019) for each watershed for baseflow periods (A) and stormflow periods (B). Watersheds are ordered by the density of drainage infrastructure from highest (*USC*) to lowest (*MRF*). Error bars show the range of loads measured over the four-year period. Insufficient data were available to estimate winter loads in *UPN* indicated with (\*).**


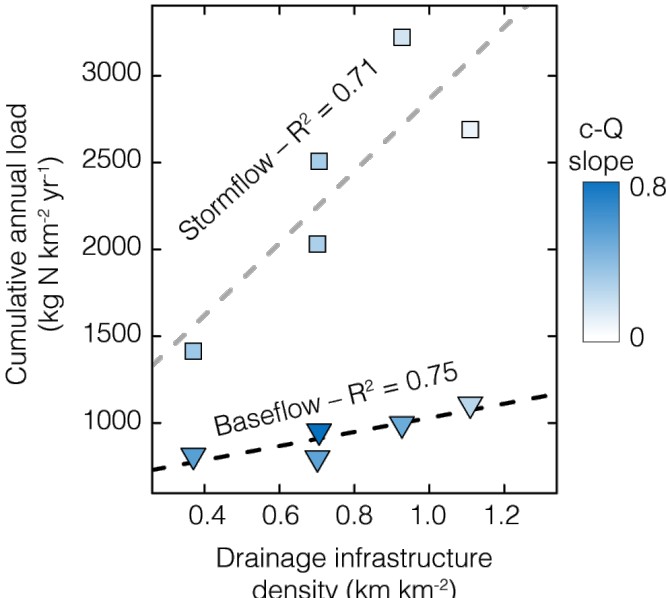

**Figure 7: For all five watersheds, cumulative annual load exported during stormflow (squares) and baseflow (triangles) periods as a function of the drainage infrastructure density. Shapes are colored by the average c-Q slope for stormflow and baseflow periods with darker blues associated with more chemodynamic export regimes.**
