# Peer review of "Hydrologic regimes drive nitrate export behavior in human impacted watersheds"

_Hydrology and Earth System Sciences, 2020_

## Referee Comment (RC1) · Anonymous Referee #1 · 10 Dec 2020

In this paper, Gorski and Zimmer examine how examine nitrate dynamics in a series of nested basins in agricultural regions of lowa. Specifically, they focus on the concentration-discharge (c-Q) relationships of nitrate in these rivers, using these relationships to make inferences about the source areas of nitrate delivered to the stream and the amount of in- and near-stream uptake that may modulate these c-Q relationships. In doing so, they divide the periods of analyses into stormflow and baseflow periods, predicated on recent work that shows that different c-Q relationships obtain in many rivers at high vs. low discharge. They find that stormflow periods were largely chemostatic (circum-zero c-Q slopes) while baseflow periods shifted between being chemodynamic and chemostatic depending on the season. They also found that in-
creased ag infrastructure drove a more chemostatic basflow response. They make several interesting and plausible hypotheses as to why this might be the case.

This is a good manuscript can be published with minor modifications. The paper provides an interesting case study of nitrate c-Q dynamics in a region that is very important for the Mississippi N budget. It is well-written and demonstrates a strong command of the extant literature. I would make the following general suggestions for improving an already-good manuscript.

- There are a LOT of comparisons made in this paper, with baseflow vs stormflow, seasons, and different metrics of land use all considered. All these comparisons, at some point, make it difficult to extract the main message from each section. I would encourage some revision to make these main points clearer.

- The authors put a lot of stock in which variable (e.g. drainage density) was the most correlated with a particular c-Q metric, and they provide some interesting hypotheses/speculation about the ultimate causes of these correlations. They sometimes neglect other variables that were nearly as well correlated as the one they were focusing on, however, and they neglect to offer alternative hypotheses that might be as plausible. Given that there are only 5 watersheds in the study, there is likely a large degree of chance that determines which variables are most correlated with c-Q dynamics, especially if there is a high degree of correlation between predictors, which I imagine there is. I think they would do well to acknowledge some of these alternative hypotheses. But...

- With the exception of drainage density, the amount of variability in LULC, both total and within set buffers, is very low, and I would urge the authors to consider whether it is plausible that such small differences might plausibly drive the differences in c-Q patterns that they observed. I don't know one way or the other, but I think it's something to consider cerefully.

Line comments:

**HESSD**
52 My opinion: this puts too much stock in the ability of c-Q relationships to identify source areas. I think that c-Q relationships can help develop hypotheses about sources, but attributing a NO3 flux to a source area by c-Q relationship alone is a very tenous thread.

110 spelling "Glycine max"

115 It is my understanding that this area was drained in the late 1800s-early 1900s. https://www.desmoinesregister.com/story/opinion/columnists/2015/03/01/story-pioneer-iowa-wetland-farmland/24212331/

132 Is this volume-weighted mean concentration or just streight mean?

135 spelling "Land Cover"

143 Why this absolute slope criterion rather than relative (either % flow change or criterion based on area-normalized Q?) Wont absolute criterion mean more storm detection in large rivers? 143 I know USGS uses cfs, but please metric-ize discharge measurements, especially since you also use L volumes in the paper. Also 1e-4cfs/s is a nonintuitive quantity (is that like a tablespoon?) and mismatched to timescale of the data. can you put in m3d-2 units instead?

194 I would remove these statistical tests for a number of reasons: 1) It's inappropriate to model count data using a t-test because a t distribution is continuous 2) you're treating watersheds as independent observations in the t-test, but there is an obvious correlation structure given that the watersheds are nested. 3) I don't think the statustics are really necessary just to say "the count of spring and summer events and their precip totals were similar."

200 "+/- 3.07" Please state once what this variability estimate is (sd? I assume not SEM)

209 "groundwater flow paths with longer residence times, and more streambed-water interaction," The relevance of these factors may not be immediately apparent to a
reader.

205-215 - Suggest reworking this paragraph a bit. The central thrust of the argument isnt entirely clear.

221 - Another good cite for conceptual framework considering this idea: Wollheim et al. 2018 Biogeochemistry

279-294 - This might suggest that nonlinear power law fitting, instead of a linear fit in log-log space, might be a more robust approach to dealing with low Q anomalies, because in these periods, absolute residuals are small but log-residuals are huge.

304 - m2 unit should be km2

344 - I'm not sure I buy this explanation... given that you have multiple other predictors that are just sliiiightly less correlated than 100-m buffer crop % I think it's a stretch to focus on just one explanation. Could also be driven by drainage density, for instance.

Supplement:

Figure S1: it looks like there is some linear interpolation going on over dates where there's no Q data e.g. MJF Dec 2017-Feb 2018. Please show gaps in Q data instead.

HESSD

---

## Referee Comment (RC2) · Douglas Burns (Referee) · 26 Dec 2020

In this study, the authors examine nitrate c-Q relationships based on high-frequency data across 5 agricultural watersheds in Iowa. They separate their data into baseflow and stormflow by applying a set of objective criteria, although they describe that some subjective decisions are necessary to finalize the data separation process. The authors focus their analysis on seasonal patterns of variation. Intensity of artificial drainage is an important explanatory variable across the watersheds, as concluded by many past studies.

Overall, this is a good study that provides some new insights to nitrate behavior in agricultural catchments, but mainly reinforces conclusions from previous investigations. The paper is well written, the methods technically sound, and the authors demonstrate good awareness of previous related papers. The limitations of a c-Q study such as this is that it's not always clear to what extent the static vs. dynamic patterns are driven by c vs. Q. In other words, much of the reason why baseflow is more chemodynamic may be that Q varies less, so that any variation in c is amplified. So, it would be helpful for the authors to comment on the relative c vs. Q roles regarding their interpretation of seasonal differences, and inter-watershed comparisons, especially relative to comparing stormflow to baseflow. I am not necessarily questioning their broad interpretations of the drivers of c-Q patterns, just asking for some additional insight as to whether c or Q are driving some of the differences described in the paper.

Beyond this criticism, the study provides limited insight to inter-annual variation. Studies of stream nitrate in the agricultural Midwest have highlighted strong year-to-year variation in c-Q patterns. For example, Jones et al., 2017 (cited by the authors) and Davis et al., 2014 (JEQ, 43: 1494-1503) provide examples of the strong role of dry periods followed by re-wetting. The authors do discuss relative wet-dry conditions on a seasonal basis, but the study provides little perspective on inter-annual patterns and the role these may have played in the study results. At least some quantitative insight would be helpful especially at it may have affected the baseflow vs. stormflow conclusions.

Below are some specifics comments and criticisms referred to line number, and these range from minor editorial suggestions to more substantive comments:

Title – should change "nutrient" to "nitrate" in title since nitrate is the only nutrient that was analyzed

Line 48 – suggest adding "excess" before applied

Line 194 – rather than refer to Student's t-test whenever paired comparison results are discussed, would be better to state this approach in the methods section and then just

describe whether results are significant or not given criteria described in methods.

Line 205 – in the paragraph that begins on this line, there is discussion of correlations, but no reference to quantitative values. It would be best to define how correlations were determined—-Pearson Product Moment or another approach? Also, should provide p value to reinforce terms such as "well correlated". Was this done just visually, or were tests performed and significance determined?

Line 291 – the authors mention the possibility of biofouling affecting nitrate concentrations during one baseflow period. Was this based on evidence from a technician that serviced the site or was it based on anomalous values or rapid unexplained shifts? Would be good to provide basis for this statement. And this does raise the question as to whether biofouling may have affected other periods of observation.

Line 323 – higher export than what? Comparative here is incomplete.

Line 375 – change "that" to "than"

Figure 3 – the color contrasts in the figure panels are not as strong as those shown in the color key. For example, the blue and green patterns for fall and winter show poor contrast in the figure. This is also true for spring in 3a and summer in 3b.

---

## Author Comment (AC1) · 20 Jan 2021

REVIEW #1 In this paper, Gorski and Zimmer examine how examine nitrate dynamics in a series of nested basins in agricultural regions of Iowa. Specifically, they focus on the concentration-discharge (c-Q) relationships of nitrate in these rivers, using these relationships to make inferences about the source areas of nitrate delivered to the stream and the amount of in- and near-stream uptake that may modulate these c-Q relationships. In doing so, they divide the periods of analyses into stormflow and baseflow periods, predicated on recent work that shows that different c-Q relationships obtain in many rivers at high vs. low discharge. They find that stormflow periods were

largely chemostatic (circum-zero c-Q slopes) while baseflow periods shifted between being chemodynamic and chemostatic depending on the season. They also found that increased ag infrastructure drove a more chemostatic basflow response. They make several interesting and plausible hypotheses as to why this might be the case. This is a good manuscript can be published with minor modifications. The paper provides an interesting case study of nitrate c-Q dynamics in a region that is very important for the Mississippi N budget. It is well-written and demonstrates a strong command of the extant literature. I would make the following general suggestions for improving an already-good manuscript.

Response: We thank the reviewer for their comments

- There are a LOT of comparisons made in this paper, with baseflow vs stormflow, seasons, and different metrics of land use all considered. All these comparisons, at some point, make it difficult to extract the main message from each section. I would encourage some revision to make these main points clearer.

Response: We thank the reviewer for this comment, we have attempted to clarify sections and make it clearer the main takeaway point from each. In addition, we have added language that was missing or unclear in some areas stating explicitly what we are comparing and why.

- The authors put a lot of stock in which variable (e.g. drainage density) was the most correlated with a particular c-Q metric, and they provide some interesting hypotheses/speculation about the ultimate causes of these correlations. They sometimes neglect other variables that were nearly as well correlated as the one they were focusing on, however, and they neglect to offer alternative hypotheses that might be as plausible. Given that there are only 5 watersheds in the study, there is likely a large degree of chance that determines which variables are most correlated with c-Q dynamics, especially if there is a high degree of correlation between predictors, which I imagine there is. I think they would do well to acknowledge some of these alternative hypotheses. But... - With the exception of drainage density, the amount of variability in LULC, both total and within set buffers, is very low, and I would urge the authors to consider whether it is plausible that such small differences might plausibly drive the differences in c-Q patterns that they observed. I don't know one way or the other, but I think it's something to consider cerefully.

Response: We agree that with the number of watersheds analyzed it is difficult to draw strong mechanistic conclusions about the drivers of c-Q patterns. However, although the differences in land use are minimal, the difference in drainage density across the watersheds is substantial (as noted by the reviewer). For this reason, as well as previous work that has been done, we hypothesize that drainage infrastructure, at least in part, drives the c-Q patterns we observe.

Line comments: 52 My opinion: this puts too much stock in the ability of c-Q relationships to identify source areas. I think that c-Q relationships can help develop hypotheses about sources, but attributing a NO3 flux to a source area by c-Q relationship alone is a very tenous thread.

Response: We suspect that the reviewer may be referencing line 55: "An effective method for investigating contributing source zones within a watershed is the examination of the relationship between solute concentration and stream discharge (c-Q relationships)" We have amended the sentence to the following: "The examination of the relationship between solute concentration and discharge (c-Q relationships), in combination with other information about watershed structure and land use practices, can be an effective way to investigate contributing source zones within a watershed."

110 spelling "Glycine max"

Response: Thank you for catching this, the spelling has been corrected

115 It is my understanding that this area was drained in the late 1800s-early 1900s. https://www.desmoinesregister.com/story/opinion/columnists/2015/03/01/storypioneer-iowa-wetland-farmland/24212331/

Response: Yes, this area has a very interesting history, and in fact some of the drainage infrastructure is from the original draining of the area is still in place. We have updated text indicate that the drainage infrastructure was installed as early as the 1800s. Thank you for this informative article.

132 Is this volume-weighted mean concentration or just streight mean?

Response: This is just a standard average (not volume-weighted), we have added text to section 2.2 to specify. Standard average was used here to facilitate comparison to maximum contaminant levels and target concentrations.

135 spelling "Land Cover"

Response: Thank you for catching this, the spelling has been corrected

143 Why this absolute slope criterion rather than relative (either % flow change or criterion based on area-normalized Q?) Wont absolute criterion mean more storm detection in large rivers?

Response: We tried several different methods for selecting events from the record, and we found that the criteria did a reasonable job, although there are likely other ways to do this. Because the watersheds are in a similar area, the hydrographs had similar structures, however, if the analysis were to be expanded to other areas with significantly different hydrograph structures a different approach such as % flow change or area normalized Q might be more appropriate. We have added language to section 2.3 to clarify this.

143 I know USGS uses cfs, but please metricize discharge measurements, especially since you also use L volumes in the paper. Also 1e-4cfs/s is a nonintuitive quantity (is that like a tablespoon?) and mismatched to timescale of the data. can you put in mË̦3dË̦-2 units instead?

Response: We understand and agree that cfs is a non-intuitive unit and we debated whether to convert the discharge measurements to metric or not. We decided against it for two reasons 1) previous studies examining these watersheds reported Q in cfs, and in an effort to facilitate comparison we kept our records in cfs and 2) we are using USGS data, which is reported in cfs, and we want to connect our work to current USGS water quality efforts.

194 I would remove these statistical tests for a number of reasons: 1) It's inappropriate to model count data using a t-test because a t distribution is continuous 2) you're treating watersheds as independent observations in the t-test, but there is an obvious correlation structure given that the watersheds are nested. 3) I don't think the statustics are really necessary just to say "the count of spring and summer events and their precip totals were similar."

Response: The reporting of t-test results has been removed.

200 "+/- 3.07" Please state once what this variability estimate is (sd? I assume not SEM)

Response: This is a standard deviation; it has been added to the text

209 "groundwater flow paths with longer residence times, and more streambed-water interaction," The relevance of these factors may not be immediately apparent to a reader.

Response: We have added the following text to make the point more apparent to the reader: "...more streambed-water interaction which would allow for more nitrate processing in the subsurface and hyporheic zone."

205-215 - Suggest reworking this paragraph a bit. The central thrust of the argument isnt entirely clear.

Response: We have reworded the paragraph to make the main point more clear which is that watersheds with more drainage infrastructure show higher NO3 concentrations

which is consistent with previous findings.

221 - Another good cite for conceptual framework considering this idea: Wollheim et al. 2018 Biogeochemistry Response: Thank you, we have added that citation

279-294 - This might suggest that nonlinear power law fitting, instead of a linear fit in log-log space, might be a more robust approach to dealing with low Q anomalies, because in these periods, absolute residuals are small but log-residuals are huge.

Response: This is an interesting idea, and we agree that c-Q relationships should not be constrained to linear fits in log-log space without careful consideration of the data and the fitting procedure.

304 - mȨ̈2 unit should be kmȨ̈2

Response: Thank you for catching that, it has been fixed.

344 - I'm not sure I buy this explanation... given that you have multiple other predictors that are just sliiiightly less correlated than 100-m buffer crop % I think it's a stretch to focus on just one explanation. Could also be driven by drainage density, for instance.

Response: We agree that 100-m buffer crop is one of many factors that might drive this behavior, however because this does offer a plausible mechanism for the trend, we submit it as a reasonable explanation. Drainage density or topology could also be a contributing factor, and we have added language to acknowledge that as well.

Supplement: Figure S1: it looks like there is some linear interpolation going on over dates where there's no Q data e.g. MJF Dec 2017-Feb 2018. Please show gaps in Q data instead.

Response: Gaps are shown in this figure and we have done no interpolation. The period that the reviewer has identified is a low flow period, similar to those discussed in section 3.5.

562, 2020.

---

## Author Comment (AC2) · 20 Jan 2021

REVIEW #2 In this study, the authors examine nitrate c-Q relationships based on high-frequency data across 5 agricultural watersheds in Iowa. They separate their data into baseflow and stormflow by applying a set of objective criteria, although they describe that some subjective decisions are necessary to finalize the data separation process. The authors focus their analysis on seasonal patterns of variation. Intensity of artificial drainage is an important explanatory variable across the watersheds, as concluded by many past studies. Overall, this is a good study that provides some new insights to nitrate behavior in agricultural catchments, but mainly reinforces conclusions from

previous investigations.

Response: While we agree that some of the manuscript's main conclusions are supported by other studies, to our knowledge there have been relatively few studies systematically examining event scale nutrient export dynamics throughout several watersheds across a broad spatial area. Existing studies have mainly focused on detailed event-based studies in a single watershed or several small watersheds or large-scale nutrient export dynamics in which the hydrograph is partitioned in a more generic way (eg. flow percentile). We have added language in the introduction and the abstract to highlight this novelty.

The paper is well written, the methods technically sound, and the authors demonstrate good awareness of previous related papers. The limitations of a c-Q study such as this is that it's not always clear to what extent the static vs. dynamic patterns are driven by c vs. Q. In other words, much of the reason why baseflow is more chemodynamic may be that Q varies less, so that any variation in c is amplified. So, it would be helpful for the authors to comment on the relative c vs. Q roles regarding their interpretation of seasonal differences, and inter-watershed comparisons, especially relative to comparing stormflow to baseflow. I am not necessarily questioning their broad interpretations of the drivers of c-Q patterns, just asking for some additional insight as to whether c or Q are driving some of the differences described in the paper.

Response: Thank you for this observation. We have added text in section 3.4 that addresses the difference between c and Q variability during baseflow and stormflow. We have also added an additional supplemental table (attached). The table shows calculations of the coefficient of variation (CV) for c and Q during baseflow and stormflow periods. Individual values of CVQ and CVc show that baseflow c-Q chemodynamic behavior is driven by both a decrease in Q variation and an increase in c variation compared to stormflow periods. The ratio of CVc:CVQ is higher during baseflow, consistent with variable sourcing of nitrate. During stormflow periods, CVc:CVQ values are lower, indicating little change in c relative to Q. These patterns are more pronounced

for the watersheds with the least amount of drainage infrastructure (UPN and MJF) than for those with more drainage. This is also consistent with our explanation for the c-Q behavior. Additionally, there is considerable overlap in c and Q values between stormflow and baseflow as shown by Figure S3. Given that baseflow and stormflow c-Q patterns differ, this suggests that partitioning of the hydrograph by events may sample different hydrologic regimes with similar discharges.

Beyond this criticism, the study provides limited insight to inter-annual variation. Studies of stream nitrate in the agricultural Midwest have highlighted strong year-to-year variation in c-Q patterns. For example, Jones et al., 2017 (cited by the authors) and Davis et al., 2014 (JEQ, 43: 1494-1503) provide examples of the strong role of dry periods followed by re-wetting. The authors do discuss relative wet-dry conditions on a seasonal basis, but the study provides little perspective on inter-annual patterns and the role these may have played in the study results. At least some quantitative insight would be helpful especially at it may have affected the baseflow vs. stormflow conclusions.

Response: We agree that year-to-year variability has been shown to have a significant effect on nutrient mobilization in these systems, however due to the length of the in-situ nitrate measurement record, this dataset cannot yet address inter-annual variability. We have added some language in section 3.5 addressing how inter-annual variability may impact the trends that we see.

Below are some specifics comments and criticisms referred to line number, and these range from minor editorial suggestions to more substantive comments:

Title – should change "nutrient" to "nitrate" in title since nitrate is the only nutrient that was analyzed

Response: We agree, and we have changed the title

Line 48 – suggest adding "excess" before applied

Response: The word has been added

Line 194 – rather than refer to Student's t-test whenever paired comparison results are discussed, would be better to state this approach in the methods section and then just describe whether results are significant or not given criteria described in methods.

Response: The results of the t-tests have been removed from the manuscript because they added little value as noted by Reviewer #1.

Line 205 – in the paragraph that begins on this line, there is discussion of correlations, but no reference to quantitative values. It would be best to define how correlations were determined—-Pearson Product Moment or another approach? Also, should provide p value to reinforce terms such as "well correlated". Was this done just visually, or were tests performed and significance determined?

Response: Quantitative values have been added to this paragraph. Language has been added to the methods section to explain how we calculated correlation, and a graphical symbol for significance has been added to Figure S4 (bold indicates p < 0.01).

Line 291 – the authors mention the possibility of biofouling affecting nitrate concentrations during one baseflow period. Was this based on evidence from a technician that serviced the site or was it based on anomalous values or rapid unexplained shifts? Would be good to provide basis for this statement. And this does raise the question as to whether biofouling may have affected other periods of observation.

Response: The basis for this statement is a rapid decrease in $[NO_3]$ and consistent values well below the rest of the record($\leq$ 0.1 percentile). This does not appear to occur, at least for an extended period of time, throughout the rest of the records or in the other watersheds, which is why we have not identified it elsewhere. We have added language to justify our conjecture.

Line 323 – higher export than what? Comparative here is incomplete.

Response: We have added language to make the comparison more clear

Line 375 – change "that" to "than"

Response: We have changed the word

Figure 3 – the color contrasts in the figure panels are not as strong as those shown in the color key. For example, the blue and green patterns for fall and winter show poor contrast in the figure. This is also true for spring in 3a and summer in 3b.

Response: We have updated the figure to make the color contrast more apparent to the reader.

| Watershed | Stormflow | | | Baseflow | | |
|---|---|---|---|---|---|---|
| | $CV_c{}^a$ | $CV_Q$ | $CV_c/CV_Q$ | $CV_c$ | $CV_Q$ | $CV_c/CV_Q$ |
| UPN | 0.412 | 1.058 | 0.390 | 0.506 | 0.603 | 0.840 |
| USC | 0.321 | 0.914 | 0.352 | 0.325 | 0.868 | 0.374 |
| MRF | 0.369 | 0.842 | 0.438 | 0.370 | 0.558 | 0.664 |
| MJF | 0.337 | 0.870 | 0.387 | 0.382 | 0.744 | 0.513 |
| DVM | 0.401 | 0.886 | 0.452 | 0.444 | 0.785 | 0.565 |

[a] Coefficient of variation = standard deviation/mean

**Fig. 1.**